# Inhibition of Amyloid Beta Aggregation and Deposition of *Cistanche tubulosa* Aqueous Extract

**DOI:** 10.3390/molecules24040687

**Published:** 2019-02-14

**Authors:** Chien-Liang Chao, Hsin-Wen Huang, Hui-Chi Huang, Hsin-Fan Chao, Shuen-Wen Yu, Muh-Hwan Su, Chao-Jih Wang, Hang-Ching Lin

**Affiliations:** 1Sinphar Pharmaceutical Co., Ltd., Sinphar Group, Yilan 269, Taiwan; chaokmc@gmail.com (C.-L.C.); lobar@sinphar.com.tw (H.-W.H.); hfchao@sinphar.com.tw (H.-F.C.); ysw@sinphar.com.tw (S.-W.Y.); smh1027@sinphar.com.tw (M.-H.S.); 2Department of Chinese Pharmaceutical Sciences and Chinese Medicine Resources, China Medical University, Taichung 404, Taiwan; hchuang@mail.cmu.edu.tw; 3School of Pharmacy, National Defense Medical Center, Taipei 114, Taiwan; 4Sinphar Tian-Li Pharmaceutical Co., Ltd., Sinphar Group, Hangzhou 311100, China

**Keywords:** *Cistanche tubulosa*, phenylethanoid glycosides, Alzheimer’s disease, iron chelation

## Abstract

*Cistanche tubulosa* aqueous extract (CTE) is already used as a botanical prescription drug for treating dementia in China. Our previous studies reported that phenylethanoid glycosides of CTE have anti-Alzheimer’s disease (AD) activity by inhibiting amyloid β peptide (Aβ) aggregation and deposition. However, recent studies considered that the phenylethanoid glycosides may be metabolized by intestinal bacteria, because all analysis results showed that the bioavailability of phenylethanoid glycosides is extremely low. In this study we demonstrate how iron chelation plays a crucial role in the Aβ aggregation and deposition inhibition mechanism of phenylethanoid glycosides of CTE. In addition, we further proved phenylethanoid glycosides (**1**–**3**) could reach brain. Active CTE component and action mechanism confirmation will be a great help for product quality control and bioavailability studies in the future. At the same time, we provide a new analysis method useful in determining phenylethanoid glycosides (**1**–**3**) in plants, foods, blood, and tissues for chemical fingerprint and pharmacokinetic research.

## 1. Introduction

Dementia is one of the most common chronic aging diseases. In 2015 the World Alzheimer Report estimated that about 46.8 million people suffered from dementia, and the number is expected to be 74.7 million in 2030 and 131.5 million in 2050 [1]. The global dementia population will increase year by year and become out of control in the future. Global healthcare expenditures to treat dementia were almost 604 billion US dollars in 2010 and the amount is expected to be 1000 billion in 2030 [2]. Dementia patients have greater risk of accidental death [3] and therefore require more medical care. This is a heavy burden for dementia patient familial caregivers. The impact of dementia on caregivers, family and society can produce great physical, psychological, life and economic stress. There are two major forms of dementia, Alzheimer’s disease (AD) and vascular dementia [4,5]. AD is the most common dementia and is an irreversible and progressive neurodegenerative disorder [4,5,6]. AD was the sixth leading cause of death in the United States in 2015 [6].

Recent studies support that the neuron toxicity induced by amyloid β peptide (Aβ) (plaques) and protein tau (tangles) aggregation are closely related to AD pathogenesis. The accumulation of Aβ and protein tau in the brain can lead to neuron damage and memory loss [7]. Insoluble Aβ oligomers aggregate in extracellular plaques and were reported to lead to synaptic dysfunction, neuron toxicity and cell death [8]. Unfortunately, there is no effective treatment available so far to clear Aβ and protein tau, or inhibit the formation or oligomerization of Aβ, and suspend or cure the irreversible neuron damage. The current therapeutic agents, such as galantamine and donepezil, which are acetyl cholinesterase inhibitors, can only temporarily ameliorate memory loss by raising the neurotransmitter level. In 2012, no candidate drugs were able to reduce the Aβ effect in large AD patient clinical studies [9]. Therefore, research for a novel treatment to suppress the prevalence of Alzheimer’s disease is urgently needed. Metal ions were recently considered a key factor closely related with Aβ aggregation [10]. Metal chelators regarded as potential lead compounds provide a whole new strategy for anti-AD approach [9,11,12,13,14,15].

The dried stem of *Cistanche tubulosa* (Schrenk) R. Wight, Rou Cong Rong, is widely harvested in the Xhinjang, China desert. It is an important traditional Chinese medicine that belongs to tonic Chinese medicine indexed in the China Pharmacopedia and used for thousands of years for the treatment of physical weakness, kidney deficiency, infertility, forgetfulness, impotence and senile constipation [16,17]. *C. tubulosa* aqueous extract (CTE) capsules have been approved as a botanical drug for vascular dementia in China. According to a clinical trial of 18 patients diagnosed with mild to moderate AD, CTE capsules gave patients a more stable memory condition compared to acetyl cholinesterase inhibitors [18]. In our previous study, the CTE decreased Amyloid β peptide (Aβ) deposition and improved memory loss in Alzheimer’s disease-like rats [19]. Phenylethanoid glycosides, echinacoside (**1**), acteoside (**2**), and isoacteoside (**3**) are considered the major components of CTE [20]. Compound **1** has neuroprotecitve activity against the toxicity of 1-methyl-4-phenyl-1,2,3,6-tetrahydropyridine (MPTP) [21], 1-methyl-4-phenylpyridinium ion (MPP+) [22], and 6-Hydroxydopamine (6-OHDA) [23]. Our previous report showed that **1** suppressed Aβ oligomerization and toxicity in an in vitro study, and inhibited Aβ deposition, improved impaired memory and cognitive dysfunction in Aβ-induced rats [24]. Compounds **2** and **3** increased Aβ degradation and decreased Aβ oligomerization in vitro [25]. Compound **3** decreased Aβ deposition in Aβ-induced rats [25]. Based on these studies, *C. tubulosa* has already proven it has the potential to treat AD. CTE is closely related to anti-Aβ aggregation and deposition [24,25]. To better clearly understand the CTE mechanism, we need to have a convenient and efficient method to identify the active component, as well as the content in brain tissue samples. Most studies were designed for quantitative determination of echinacoside, acteoside, and isoacteoside in herbs or foods by high performance liquid chromatography (HPLC) [26,27,28], but only two studies were for determination in serum or tissues [29,30]. However, there is no report on the simultaneous analysis of **1**–**3** in brain tissues after CTE oral administration. In addition, it is still unclear from recent studies whether the CTE active components or their metabolites successfully passed through the blood-brain barrier and acted on brain neurons. Therefore, this research provides an advanced and sensitive method with convincing evidence that clearly illustrates the underlying mechanism of CTE against Aβ aggregation and deposition.

## 2. Results

### 2.1. Isolation and Identification of Three Major Compounds (**1**–**3**)

Dried stems of *C. tubulosa* were extracted with 75% EtOH at room temperature. The condensed 75% EtOH extract was subjected to resin and C18 column chromatography to furnish three major compounds, echinacoside (**1**), acteoside (**2**) and isoacteoside (**3**). Their structures were elucidated by analysis of the NMR (Nuclear Magnetic Resonance) spectroscopy and ESI-MS (Electrospray ionization mass) and by comparison with literature data [31,32,33] (Figure 1, Appendix A).

### 2.2. The Phenylethanoid Glycosides (**1**–**3**) Content Assay from CTE

The assay method was validated for phenylethanoid glycosides analysis. Compounds **1**–**3** (5.0 mg) were accurately weighed and mixed with 50% methanol (40 mL). The mixtures were sonicated for 5 min. Three phenylethanoid glycosides (**1**–**3**) standard solution was mixed as the standard stock solution (100 μg/mL). The standard stock solution was used to prepare five different concentrations (5, 10, 25, 50, and 100 μg/mL) of the indicated solutions, respectively, by diluting with serial volumes of 50% methanol. The phenylethanoid glycosides (**1**–**3**) curves were established using ultra performance liquid chromatography (UPLC) from 5–100 μg/mL and were linear respectively (**1**, *r*^2^ = 0.9999; **2**, *r*^2^ = 0.9999; **3**, *r*^2^ = 0.9998). The intra-day and inter-day precision [% R.S.D. (relative standard deviation) was from 0.1 to 1.8%] and accuracy (95.6–104.2%) were acceptable by UPLC for **1**–**3**. UPLC quantified the **1**, **2**, and **3** of CTE contents to be 254, 38, and 41 mg/g, respectively (Appendix A). 

### 2.3. UPLC/MS/MS Validation

The phenylethanoid glycosides (**1**–**3**) were mixed and spiked with rat brain tissue homogenate and analyzed using Ultra performance liquid chromatography-tandem mass spectrometer (UPLC/MS/MS) with solid phase extraction. The lower limit of detection (LLOQ) of **1**–**3** was 0.2 ng/mL. The phenylethanoid glycosides recovery spiked with brain tissue homogenate (10 ng/mL) at 91.39% for **1**, 89.54% for **2**, and 96.81% for **3**, respectively (Figure 2).

### 2.4. Analysis of AD-Like Rat Brain Tissues

After 14 days of CTE oral administration (200 mg/kg/day), AD-like rat brain tissues were collected and divided into two parts, hippocampus and striatum, and were both homogenized [19]. Because the hippocampus and striatum homogenates from each rat were too scant to be loaded into solid phase extraction and to avoid excessive experimental errors, the hippocampus and striatum homogenates from four individual rats were mixed into an analyzed sample for UPLC/MS/MS using the solid phase extraction method (Figure 3A,B). The phenylethanoid glycosides (**1**–**3**) were observed in the hippocampus (Figure 3A) and striatum (Figure 3B). There was no significant peak in blank brain tissues (Figure 3C). In the hippocampus the echinacoside content was 11.97 ± 0.34 ng/mL, acteoside content was 1.25 ± 0.16 ng/mL, and isoacteoside content was 1.38 ± 0.08 ng/mL. In the striatum the echinacoside content was 22.60 ± 1.69 ng/mL, acteoside content was 2.03 ± 0.61 ng/mL, and isoacteoside content was 4.90 ± 0.64 ng/mL. As shown in Figure 4, the CTE would pass through the blood brain barrier according to the significant detectable amounts of **1**–**3** in the hippocampus and striatum. The content of **1** is much higher than **2** and **3** in CTE by UPLC analysis (Appendix A). Therefore, it is reasonable to observe that **1** is the highest in brain tissue after oral CTE administration.

### 2.5. The Phenylethanoid Glycosides (**1**–**3**) Metal Chelating Activity.

In the literature, the determination of the free-from compounds and their metal complex were performed using HPLC [34,35]. Measure of chelation activity were analyzed by UPLC in this study. The 50 μg/mL solutions of each the phenylethanoid glycosides (**1**–**3**) were added into a 10 μg/mL solution of copper (Cu), calcium (Ca), magnesium (Mg), zinc (Zn), iron (Fe) or rat serum respectively. Each compound with each metal solution or serum was analyzed by UPLC. The phenylethanoid glycosides (**1**–**3**) exhibted metal chelating activity with iron and serum (Figure 5, Appendix A).

### 2.6. Anslysis of Echinacoside (**1**) in Rat Serum in Vivo

In order to understand the distribuion of echinacoside in rat serum, the serum was collected from the initial time to 720 min after echinacoside (**1**) oral administration (100 mg/kg) and analyzed by UPLC. Figure 6 showed the concentration versus time profiles of echinacoside in rat serum.

## 3. Discussion

Dementia, especially AD, is an irreversible aging and chronic disease that leads to a Gordian knot of questions. This insurmountable disease leads to serious economic issues with a huge and increasing financial burden. Worldwide, AD researchers take lots of effort to investigate new anti-AD drugs, but failed in several large clinical trials targeting Aβ in 2012 [9]. Except for directly targeting Aβ, metal ion homeostasis in the brain is considered the key reason related to Aβ aggregation and deposition which leads to AD formation [9,10]. Therefore, metal ions play an essential role in the pathogenesis of AD, and the metal chelation hypothesis has become an important research direction [9,13,14,15].

According to our previous studies [18,19,24,25], CTE displayed potential anti-dementia activity. Even in human clinical studies, after 1 year of treatment with CTE capsules for moderate AD patients, the Alzheimer’s Disease Assessment Scale-cognitive subscale (ADAS-cog) score would not show significant deterioration compared to before treatment [18]. CTE kept the ADAS-cog score of AD patients stable, but other prescription drugs did not. The only choice of prescription drugs licensed for AD treatment is acetylcholinesterase inhibitors (AChEIs) such as donepezil and galanthamine that improve AD symptoms in the short term, but deterioration occurs after 1 year of treatment [36]. Based on clinical studies, CTE has the opportunity to be a potential treatment for anti-AD agent development. Based on our previous reports [24,25], echinacoside (**1**), acteoside (**2**), and isoacteoside (**3**) would protect neurons from damage by Aβ, decrease Aβ oligomerization in vitro, and significantly ameliorate cognitive dysfunction induced by Aβ in vivo. The in vitro study indicated the active doses of **1**–**3** were up to 50 μg/mL [24,25]. But previous studies considered that the oral bioavailability of **1** and **2** were very low (0.83% for **1** and 0.12% for **2**) in rats [29,30]. Compound **1** could not even be identified in human serum after oral echinacea tablet administration [37]. Current studies show that **1**–**3** had poor membrane permeability and absorption in intestinal cells [38] and most of the **1** would be metabolized in gastrointestinal ducts [39]. Compound **2** content in rat serum was only about 4.5 μg/mL after 15 min of intravenous injection with a dose of 10 mg/kg of acteoside [30]. However, this study shows that **1**–**3** would be detected in the hippocampus and striatum of rat brain tissues after CTE oral administration (Figure 3 and Figure 4). In addition, Figure 6 showed that compound **1** content in rat serum increased about 5 times from the 15 min to 720 min after compound **1** oral administration (100 mg/kg). We considered phenylethanoid glycosides of CTE would pass through the blood brain barrier and there would be chelation action between CTE and the metal. Chelation is a reversible chemical reaction. Iron is the essential element in serum and the brain [10,40]. Figure 5 shows that three phenylethanoid glycosides would have metal chelating activity with iron and serum. Iron and serum obviously change the peak retention time area of three phenylethanoid glycosides. Iron chelation may be the crucial reason leading to misidentification of the very low bioavailability of phenylethanoid glycosides. Therefore, the metal chelating activity should be considered in blood serum and brain analysis of the three CTE phenylethanoid glycosides to recover the real bioavailability and pharmacokinetics. In addition, the results indicated three phenylethanoid glycosides (**1**–**3**) would pass through the blood brain barrier and arrived at brain tissues through body circulation. This study developed a most rapid and sensitive method for **1**–**3** analysis using UPLC/MS/MS and provided powerful evidence to prove that phenylethanoid glycosides (**1**–**3**) are the major bioactive constituents of CTE.

## 4. Materials and Methods

### 4.1. Plant Material

Stems of *C. tubulosa* were collected in May 2016 from Hangzhou, China. The plant was identified by Dr. Lin H.C. Voucher specimens (No. 20016CT) have been deposited at Sinphar Tian-Li Pharmaceutical Co., Ltd., Hangzhou, China.

### 4.2. Isolation and Purification

The dried *C. tubulosa* stem was ground into powder, and then extracted five times with 75% EtOH. After solvent evaporation under reduced pressure, the crude extract was subjected to Macroporous resin AB-8 column chromatography with H_2_O/EtOH gradient solvent systems from 20% EtOH up to 100% EtOH. According to the thin layer chromatography, four fractions (Fr.1~Fr.4) were collected for further separation. Fr. 2 was subjected to preparative high performance liquid chromatography (HPLC) on a COSMOSIL^®^ 5C18-AR-II column (250 mm × 20 mm i.d., 5 μm) using 18% acetonitrile as the mobile phase system. The flow rate was 15 mL/min. Three major peaks of interest were selectively collected. The fractions containing the targeted compounds were further condensed to dryness and produced **1**, **2**, and **3**, respectively.

### 4.3. Experimental Analysis

^1^H and ^13^C NMR spectra were obtained using Bruker Avance DRX 500 MHz spectrometers (Billerica, MA, USA) with tetramethylsilane (TMS) as the internal standard. Preparative HPLC was performed using a reverse phase column (Cosmosil C^18^-AR-II column, 250 mm × 20 mm i.d.; Nacalai Tesque, Inc., Kyoto, Japan) on a Shimadzu LC-6AD series apparatus with Prominence HPLC UV-Vis detectors (Kyoto, Japan). 

Echinacoside (**1**): white powder; ESI−MS *m*/*z*: 785 [M − H]^−^; ^1^H NMR (CD_3_OD, 500 MHz): δ 1.07 (1H, d, *J* = 6.2 Hz, Rha-H-6), 2.79 (2H, t, *J* = 5.2 Hz, H-7), 3.56 (1H, m, Ha-8), 3.79 (1H, m, H-3′), 3.91 (1H, m, Hb-8), 4.29 (1H, d, *J* = 7.7, Glc-H-1), 4.38 (1H, d, *J* = 7.9 Hz, H-1′), 5.00 (1H, t, *J* = 9.6 Hz, H-4′), 5.17(1H, d, *J* = 1.5 Hz, Rha-H-1), 6.27 (1H, d, *J* = 15.9 Hz, H-8″), 6.58 (1H, dd, *J* = 8.1,2.0 Hz, H-6), 6.67 (1H, d, *J* = 8.1 Hz, H-5), 6.70 (1H, d, *J* = 2.0 Hz, H-2), 6.77 (1H, d, *J* = 8.2 Hz, H-5″), 6.94 (1H, dd, *J* = 8.3, 2.0 Hz, H-6″), 7.05 (1H, d, *J* = 2.0 Hz, H-2″), δ 7.59 (1H, d, *J* = 15.9 Hz, H-7″); ^13^C-NMR (CD_3_OD, 125 MHz): δ 18.5 (Rha-C-6), 36.6 (C-7), 62.6 (Glc-C-6), 69.4 (C-6′), 70.5 (C-4′), 70.6 (Rha-C-5), 71.5 (Glc-C-4), 72.0 (Rha-C-3), 72.3 (C-8), 72.4 (Rha-C-2), 73.8 (Rha-C-4), 74.7 (C-5′), 75.1 (Glc-C-2), 76.1 (C-2′), 77.8 (Glc-C-5), 77.9 (Glc-C-3), 81.7 (C-3′), 103.1 (Rha-C-1), 104.2 (C-1′), 104.7 (Glc-C-1), 114.7 (C-2″), 115.3 (C-8″), 116.3 (C-5″), 116.5 (C-2), 117.1 (C-5), 121.3 (C-6), 123.3 (C-6″), 127.6 (C-1″), 131.5 (C-1), 144.7 (C-3), 146.1 (C-4), 146.9 (C-4″), 148.2 (C-7″), 149.8 (C-3″), 168.5 (C-9″) (Appendix A).

Acteoside (**2**): white powder; ESI−MS *m*/*z*: 623 [M − H]^−^; ^1^H NMR (DMSO-*d6*, 500 MHz): δ 0.94 (3H, d, *J* =5.9 Hz, H-6‴), 2.68 (2H, m, H-7), 4.33 (1H, d, *J* = 7.8 Hz, H-1″), 4.70 (1H, t, *J* = 9.0 Hz, H-4″), 5.01 (1H, s, H-1‴), 6.18 (1H, d, *J* = 15.9 Hz, H-8′), 6.48 (1H, dd, *J* = 7.9, 1.4 Hz, H-6), 6.62 (1H, d, *J* = 7.1 Hz, H-5), 6.73 (1H, d, *J* = 2.0 Hz, H-2), 6.75 (1H, d, *J* = 7.9 Hz, H-5′), 6.97 (1H, d, *J* = 8.1 Hz, H-6′), 7.01 (1H, s, H-2′), 7.44 (1H, d, *J* = 15.8 Hz, H-7′); ^13^C NMR: (DMSO-*d*6, 125 MHz): δ 18.2 (C-6‴), 35.1 (C-7), 60.8 (C-6″), 68.8 (C-5‴), 69.2 (C-4″), 70.3 (C-3‴), 70.3 (C-8), 70.4 (C-2‴), 70.6 (C-4‴), 71.7 (C-5″), 74.5 (C-2″), 79.2 (C-3″), 101.3 (C-1‴), 102.3 (C-1″), 113.6 (C-8′), 114.7 (C-2′), 115.5 (C-5), 115.8 (C-5′), 116.3 (C-2), 119.6 (C-6), 121.5 (C-6′), 125.6 (C-1′), 129.2 (C-1), 143.6 (C-4), 145.0 (C-3), 145.0 (C-3′), 145.6 (C-7′), 148.5 (C-4′), 165.7 (C-9′) (Appendix A).

Isoacteoside (**3**): white powder; ESI−MS *m*/*z*: 623 [M − H]^−^; ^1^H NMR (DMSO-*d6*, 500 MHz): δ 1.07 (3H, d, *J* = 6.1 Hz, H-6‴), 2.65 (2H, m, H-7), 4.35 (1H, d, *J* = 10.0 Hz, H-1″), 5.10 (1H, s, H-1‴), 6.27 (1H, d, *J* = 15.9 Hz, H-8′), 6.45 (1H, dd, *J* = 8.0, 1.8 Hz, H-6), 6.58 (1H, d, *J* = 8.8 Hz, H-5), 6.73 (1H, d, *J* = 1.6 Hz, H-2), 6.74 (1H, d, *J* = 8.1 Hz, H-5′), 6.94 (1H, d, *J* = 8.2 Hz, H-6′), 7.04 (1H, s, H-2′), 7.46 (1H, d, *J* = 15.8 Hz, H-7′); ^13^C NMR (DMSO-d6, 125 MHz): δ 17.9 (C-6‴), 35.2 (C-7), 63.5 (C-6″), 68.2 (C-5‴), 70.4 (C-4″), 70.6 (C-3‴), 71.7 (C-8), 72.1 (C-2‴), 73.7 (C-4‴), 74.1 (C-5″), 74.5 (C-2″), 80.9 (C-3″), 100.7 (C-1‴), 102.7 (C-1″), 113.9 (C-2′), 114.9 (C-8′), 115.5 (C-5), 115.8 (C-5′), 116.3 (C-2), 119.6 (C-6), 121.5 (C-6′), 125.5 (C-1′), 129.2 (C-1), 143.5 (C-4), 145.0 (C-3), 145.3 (C-3′), 145.6 (C-7′), 148.5 (C-4′), 165.6 (C-9′) (Appendix A).

### 4.4. Preparation of C. Tubulosa Aqueous Extract (CTE)

Stem of *C. tubulosa* powder was extracted twice by refluxing with water for 1.5 h, and the extract solution was filtered. The filtered aqueous extract (CTE) was stored at 4 °C before use.

### 4.5. Animal Model and Experimental Schedule

The AD-like rat model and experimental schedule used in this study were as prescribed in our previous study [19]. After infusing Aβ 1–42 (300 pmole/day) into the cerebral ventricle for at least two weeks or more, the AD-like rats were given CTE (200 mg/mL) by oral administration on the next day. After 14 days of administration, rats were sacrificed to collect the brain tissues. To avoid excessive experimental errors, samples from four individual rats were combined into one sample. There were three samples for each group.

### 4.6. Chemicals and Reagents

Methanol (HPLC grade) was purchased from Merck (Darmstadt, Germany). Formic acid (98% for analysis, ACS) was purchased from Panreac (Barcelona, Spain). Acetonitrile (HPLC grade) was purchased from J.T. Baker (Phillipsburg, NJ, USA). Pure water was prepared using the ELGA system (Woodridge, UK). Hesperidin was purchased from Chromadex and used as internal standard (IS) for UPLC analysis.

### 4.7. UPLC/MS/MS Apparatus

Waters Acquity TQD ultra performance liquid chromatography (UPLC)/MS/MS System (Milford, MA, USA) was used for UPLC/MS/MS analysis. The chromatographic experiments were conducted on an Acquity UPLC BEH C18 column (2.1 × 100 mm, 1.7 μm, Waters) under gradient elution at 0.4 mL/min flow rate. The mobile phase was composed of formic acid-water (1: 1000, *v*/*v*) (A) and formic acid-acetonitrile (1: 1000, *v*/*v*) (B). The time program was as follows: 0–5 min, A–B (92:8, *v*/*v*) changed to A–B (62:38, *v*/*v*); 5–7 min, A-B (62:38, *v*/*v*); 7-7.1 min, A-B (62:38, *v*/*v*) changed to A–B (92: 8, *v*/*v*); 7.1–9 min, (92: 8, *v*/*v*). The injection volume was 5 μL into the UPLC/MS/MS system with a negative electrospray ionization (ESI) ion source. The total ion chromatograms (TIC) for **1**–**3** were carried out using the multiple-reaction monitoring (MRM) mode. The mass parameters were optimized to obtain the clearest, high intensity signals. The best choice of parameters were described as 3000 V capillary voltage, 20 V cone voltage, 125 °C source temperature, 400 °C desolvation temperature, 50 L/h cone gas flow, 900 L/h desolvation gas flow, and 38 eV collision energy. Data was processed by the MassLynx 4.1 (Milford, MA, USA) software.

### 4.8. Sample Preparation

Rat brain tissue used 50% methanol for homogenization (1 g of tissue for 5 mL of methanol). The homogenate centrifuged at 10,000 rpm at 4 °C for 10 min. The supernatant (50 μL) with 10 μL IS (100 μg/mL) added pure water to 1 mL. The mixture was carefully loaded onto the Oasis HLB (hydrophilic-lipophilic balanced) cartridge (Waters, Milford, MA, USA), which had already been eluted with 1mL of methanol and then equilibrated with 1 mL of water under vacuum conditions. After 1 mL of 10% methanol was used for washing the cartridge, 1 mL of methanol was used to elute to obtain the analyzed sample which was transferred into another 1.5 mL Eppendorf tube with 10 μL of 1% ascorbic acid solution and dried by evaporation. Finally, the residue from the sample analyzed was dissolved in 200 μL of 50% methanol for analysis.

### 4.9. Preparation of the Calibration Standard Curve

Compounds **1**–**3** were used to prepare calibration standards respectively added to brain tissue homogenate and hesperidin as internal standard to obtain the final calibration standard concentrations which were 1.0, 5.0, 10.0, 50.0, and 100.0 ng/mL.

### 4.10. Validation

The calibration standard curve was obtained using the linearity test using five different concentrations of **1**–**3**, in the ranges 1.0, 5.0, 10.0, 50.0, and 100.0 ng/mL, respectively, and was regarded as linear with over 0.99 of coefficient of determination (R-squared). The lower limit of detection (LLOQ), defined as ± 20% of the concentration of a test sample, produced a signal peak rather than noise. The credibility and accuracy of the analysis method was evaluated using a recovery test at 10 ng/mL of **1**–**3**.

### 4.11. Metal Chelating Activity Assay

UPLC was used to analyze the metal chelating effect using the 50 μg/mL solution of CTE phenylethanoid glycosides **1**–**3** containing 10 μg/mL of metal solution of copper, calcium, magnesium, zinc, iron or rat serum. Each compound with each metal solution or serum was analyzed by UPLC. The metal chelating effect would change the retention time for **1**–**3** in liquid chromatography. The percentage of metal chelating effect was calculated as below and the data represented mean ± S.D. (*n* = 3).
(1)$$\mathrm{Metal}\;\mathrm{chelating}\;\mathrm{effect}\;(\%)\;=\;100-\;\left\{{\left[\frac{\mathrm{Phenylethanoid}\;\mathrm{glycosidespiked}\;\mathrm{metal}\;\mathrm{or}\;\mathrm{serum}}{\mathrm{Phenylethanoid}\;\mathrm{glycosidespiked}\;\mathrm{blank}}\right]}_{\;\mathrm{peak}\;\mathrm{area}\;\mathrm{ratio}}\times 100\right\}$$

## 5. Conclusions

This study demonstrated that **1**–**3** are the active components of CTE for anti-AD activity. Iron chelation has become the new concept for designing a new generation of drugs for the treatment of AD [41]. CTE is a potential botanical anti-AD Chinese medicine targeting iron chelation induced Aβ aggregation and deposition. In addition, we further proved that **1**–**3** would reach the brain though blood–brain barrier (BBB). Active components and underlying mechanism confirmation of CTE will greatly help quality control and the bioavailability of product studies in the future. At the same time, the advanced and efficient analysis method presented will be useful in determining **1**–**3** in plants, foods, blood, and brain tissues for chemical fingerprint and pharmacokinetic research.

## Figures and Tables

**Figure 1 molecules-24-00687-f001:**
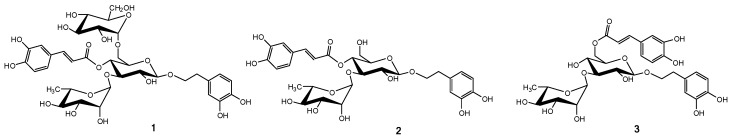
Structures of **1**, **2**, and **3** isolated from *C. tubulosa*.

**Figure 2 molecules-24-00687-f002:**
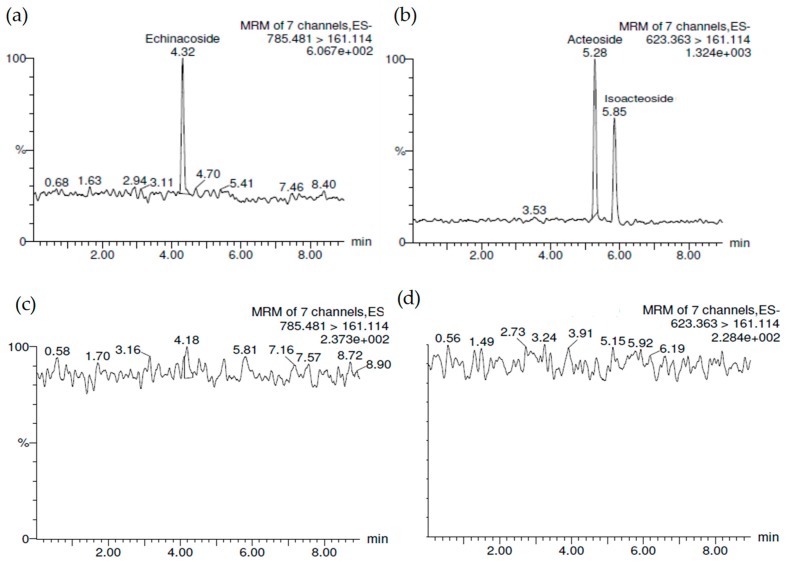
The UPLC/MS/MS chromatograms of (**a**) **1** (*t*_R_ = 4.32 min) and (**b**) **2** (*t*_R_ = 5.28 min) and **3** (*t*_R_ = 5.85 min). (**c**,**d**) Blank brain tissue. The phenylethanoid glycosides (**1**–**3**) were respectively spiked with rat brain tissue homogenate.

**Figure 3 molecules-24-00687-f003:**
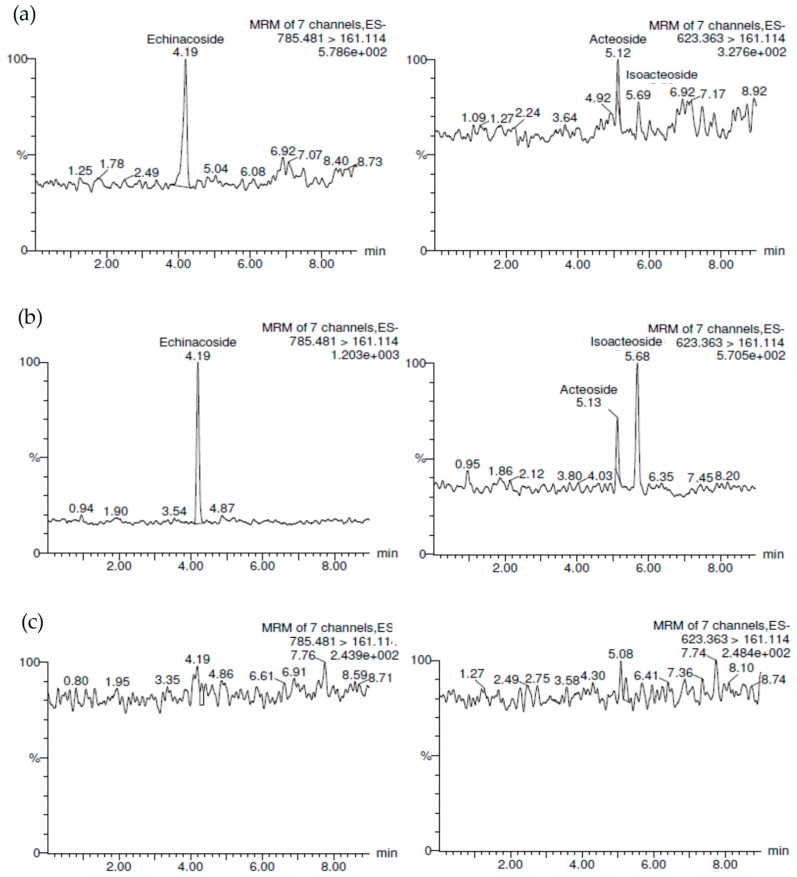
The UPLC/MS/MS chromaotograms for phenylethanoid glycosides (**1**–**3**). (**a**) hippocampus samples from Alzheimer’s disease (AD)-like rats after *Cistanche tubulosa* aqueous extract (CTE) oral administration (200 mg/kg, p.o.) [**1** (*t*_R_ = 4.19 min), **2** (*t*_R_ = 5.12 min) and **3** (*t*_R_ = 5.69 min)]. (**b**) Striatum sample from AD-like rats after CTE oral administration (200 mg/kg, p.o.) [**1** (*t*_R_ = 4.19 min), **2** (*t*_R_ = 5.13 min) and **3** (*t*_R_ = 5.68 min)]. (**c**) Blank brain tissue.

**Figure 4 molecules-24-00687-f004:**
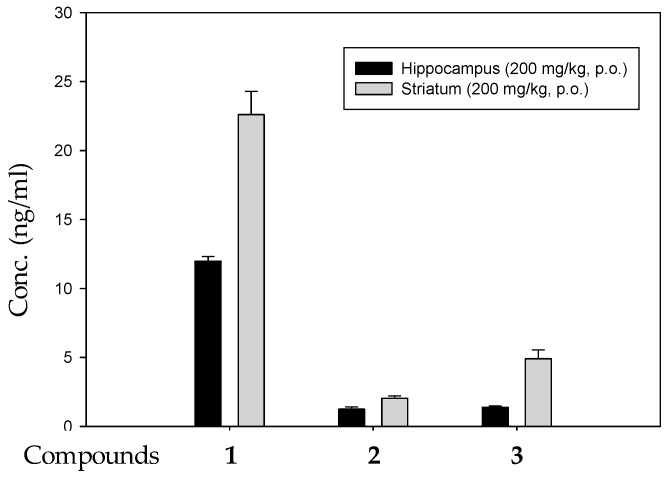
The phenylethanoid glycosides (**1**–**3**) were detectable in rat brain tissues [19] using UPLC/MS/MS. The data represented mean ± S.D., *n* = 3 for each group.

**Figure 5 molecules-24-00687-f005:**
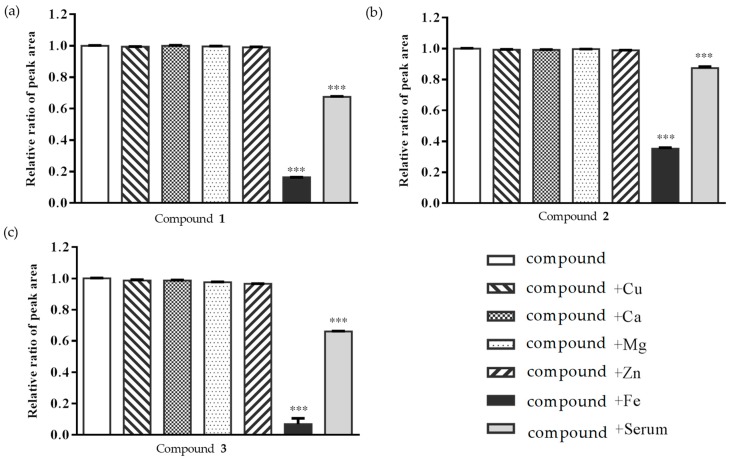
Metal chelating activity analysis. Measure of chelation activity retained in plasma for the phenylethanoid glycosides, (**a**) compound **1**, (**b**) compound **2** and (**c**) compound **3** in the presence of copper (Cu), calcium (Ca), magnesium (Mg), zinc (Zn), iron (Fe), or rat serum by UPLC analysis. The data represented mean ± S.D., *n* = 3. *** *p* < 0.001 compared with compound alone.

**Figure 6 molecules-24-00687-f006:**
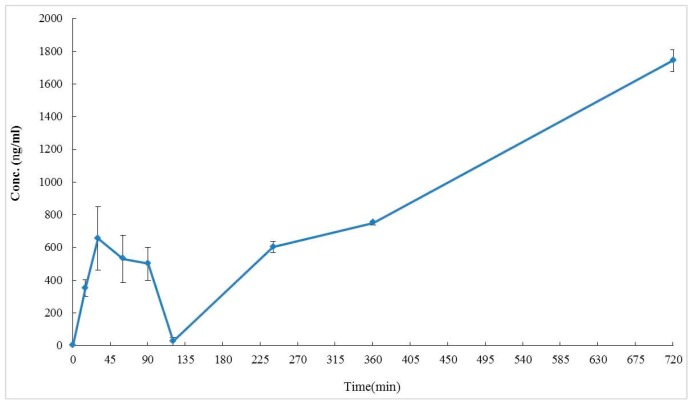
Concentration-time profile of **1** in rat serum after **1** oral administration (100 mg/kg). The data represented mean ± S.D., *n* = 3.

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
