# Peer review of "Inhibition of Amyloid Beta Aggregation and Deposition of Cistanche tubulosa Aqueous Extract"

_molecules, 2019, doi:10.3390/molecules24040687_

Round 1

Reviewer 1 Report

In the discussion part, page 6 and lines 179-182, the authors supported their conclusion by unpublished in-vivo study. This study should be included in this manuscript in order to give the reader the ability to judge its validity. Otherwise, I recommend to revise the discussion and conclusion parts accordingly. 

Author Response

Response to Reviewers

We thank the reviewers for their constructive and helpful suggestions; we have provided our responses to the reviewers’ comments and believe our manuscript is much improved as a result. This manuscript was also edited by Joel A. Newson, English Editor and Technical Writer for English grammar and syntax. All the revised portions were marked in red bold.

Response to Reviewer 1 Comments

1. In the discussion part, page 6 and lines 179-182, the authors supported their conclusion by unpublished in-vivo study. This study should be included in this manuscript in order to give the reader the ability to judge its validity. Otherwise, I recommend to revise the discussion and conclusion parts accordingly

Response 1: In lines 158-164, and 193-195, we supported the conclusion by unpublished in-vivo study as “2.6. Anslysis of Echinacoside (1) in Rat Serum in Vivo. In order to understand the distribuion of echinacoside in rat serum, the serum were collected from initial time to 720 min after echinacoside oral administration (100 mg/kg) and analyzed by UPLC. Figure 6 showed the concentration versus time profiles of echinacoside in rat serum.” and “In addition, figure 6 showed that compound 1 content in rat serum increased about 5 times from the 15 min to 720 min after compound 1 oral administration (100 mg/kg).in the revised manuscript.

Reviewer 2 Report

 The manuscript by Chao et al., investigates the blood brain barrier permeability and metal chelating possibilities of three bioavailable compounds in Cistanche tubulosa aqueous extract (CTE). This study is based on previous studies that discovered the potency of CTE in inhibiting Ab aggregation, which is linked to the Alzheimer’s disease. Previous investigations suggested poor bioavailability of CTE compounds, but the present study with UPLC/MS/MS technique was in correspondence with the results in reference 24, showing the 1, 2 and 3 CTE components are blood brain barrier permeable. Additionally, iron chelation for CTE components was observed, that may be linked to anti- Ab-aggregation potency of CTE.

UPLC/MS/MS technique is a precise method to detect different molecule contents, and it is a judicious choice to analyse CTE content in brain cells. Since the presence of CTE in brain is discordant to the previous studies, and as the main finding of this article is detecting CTE in rat brain cells, a more robust reasoning for the failure of the previous studies (instead of linking it to metal chelation in discussion) would be beneficiary. Moreover, it is known over a decade that iron is associated with Ab aggregation. Hence, detecting iron chelating properties for CTE is an auspicious finding. Accordingly, in abstract, the second finding is conveyed to be demonstrating the link between iron chelation and aggregation of Ab in the presence of CTE. However, the article lacks to provide compelling proof whether iron chelating of CTE is associated with effect of CTE on Ab aggregation. One might be misled to think that Ab aggregation inhibitory effects of CTE is specifically related to metal chelation, while the claim is only in the form of a theory. Additionally, it is known that iron is usually in the form of heme, rather than in the form of free ions. The significance of the findings on iron chelating effect of CTE may be questioned.

   In line 28, I did not find compelling evidence for the sentence in line 28:  “we demonstrate how iron chelation plays a crucial role in the Aβ aggregation and deposition inhibition mechanism of phenylethanoid glycosides of CTE.” (The article points out that the retention time was decreased, but no experiment was performed to see if CTE components change the Ab aggregation process or even the inhibition. As a result, the presented remark is misleading.)

  In line 30, I could not find in the manuscript anything related to this sentence: “We further proved how phenylethanoid glycosides (1–3) reach brain.”  The glycosides were discovered in the rat brains, but how they reach brain is not investigated.

As the main finding of this article is detecting CTE in rat brain cells, and to make the article more interesting, a more robust discussion is recommended on why the previous studies failed to observe CTE in brain cells

In line 42: “US dollars 604 billion in 2010” should be changed to 604 billion US dollars.

In line 79, provide a reference for sentence: CTE is closely related with anti-Aβ deposition and metal chelation.

In line 83, what do you mean by “rare”? The sentence needs to be reformatted.

In line 88, should be corrected into “underlying mechanism of CTE against Aβ aggregation”. CTE   aggregation implies that Ab aggregation is present even in presence of CTE.

In line 102, please clarify if all three compounds were mixed with 50% Methanol or each compound was mixed with 50% Methanol?

In line 103, according to previous comment, if there were three standard solutions, glycoside should not be plural in “Three phenylethanoid glycosides (13) standard solutions …”

In line 105, what does the word “respectively” correspond to?

The verb in the sentence in Line 108 is missing: The intra-day and inter-day precision (% R.S.D. was from 0.1 to 1.8%) and accuracy  (95.6–104.2%) by UPLC for 1–3.
In Figure 2,  indicate what the unnamed peak stands for.

In Figures 2-5: In all UPLC experiments, the control is missing and must be provided.

The quality of the picture in Figure 2 is better to be enhanced. Figure 2, could be moved to supplementary information.

Line 124: Explanation for Figures. 4A and 4B is missing in the main text. Explain the significance of these chromatograms.

Figure 5, how was obtained? If it is explained in ref 19, then needs to be referenced.

Line 143: The sentence following is unclear: The assay was based on that metal chelation changes the retention time of free-form compounds.

Line 144: The comment1-3 were prepared 50 μg/ml” should be rephrased.

Line 152: The sentenceThe data represented mean ± S.E., n=3. ***P <0.001.” is unclear. What is P?

For Figure 6, explain how the error estimates were acquired. Maybe it is better to insert this answer in the method section.

Line 156: The comment:but failed in large clinical trials targeting Aβ of 2012”, should be rephrased. If 2012, is referred to the year, it should be noted that there have been much more clinical trials since 2012, that they are worth mentioning, although most of them have failed.

Line 160: “metal chelation hypothesis has become the newest important research direction.” I have to disagree with this comment. The metal chelation in Ab was discovered back more than 2 decades ago. And there have been a lot of studies targeting metal chelation, in the past few years. Many research groups have targeted tau or Ab with antibodies, which are used in the most recent clinical trials.  Hence, the word “newest” should be removed from the sentence.

Line 178: The following remark is not convincing: However, this study shows that 13 would be detected in the hippocampus and striatum of rat brain tissues after CTE oral administration (Figure 5). In Figure 5, the concentration of 2 and 3, especially for 2 are very small. This is in correspondence of the line 177 remark, according to reference 30. The importance of much higher concentration of compound 1, compared to 2 should be emphasized.

Line 185: Please explain why, or provide a reference for the following comment: The iron metal chelation is the crucial reason leading to misidentification of the very low bioavailability of phenylethanoid glycosides.

Line 187: The sentence: “Figure 6 shows that three phenylethanoid glycosides would have metal chelating activity with iron and serum respectively.” might not be true, as there is possibility that only one or two of the compounds is reacting with iron and serum. Hence, a further study is required in which these compounds are separated and then the experiment is repeated for each compound. The same remark is applicable for the next two sentences (lines 185-190).

Line 188: What are you referring by using the word “respectively”?

Line 250: at 4 (the unit is missing!)

Line 254: explain how you performed “infusing Aβ 1-42 into the cerebral ventricle” How much Aβ was infused?

Figure 3 and 4, why the graph for echinacoside is separate from the other two compounds?

Line 307: rephrase the sentence, I believe you meant the 50 μg/ml solutions of 1-3 were prepared, containing …”

Line 308: It is unclear what the word “respectively” stands for.

Author Response

Response to Reviewers

We thank the reviewers for their constructive and helpful suggestions; we have provided our responses to the reviewers’ comments and believe our manuscript is much improved as a result. This manuscript was also edited by Joel A. Newson, English Editor and Technical Writer for English grammar and syntax. All the revised portions were marked in red bold.

Response to Reviewer 2 Comments

In line 28, I did not find compelling evidence for the sentence in line 28: “we demonstrate how iron chelation plays a crucial role in the Aβ aggregation and deposition inhibition mechanism of phenylethanoid glycosides of CTE.” (The article points out that the retention time was decreased, but no experiment was performed to see if CTE components change the Ab aggregation process or even the inhibition. As a result, the presented remark is misleading.)

Response 1: In lines 26-27, it has been added as “We considered that the actual bioavailability of phenylethanoid glycosides is underestimated based on our previous studies which indicated the active doses of echinacoside (1), acteoside (2), and isoacteoside (3) were up to 50 μg/ml to inhibit Aβ aggregation and deposition in vitro.”in the revised abstract.

 In line 30, I could not find in the manuscript anything related to this sentence: “We further proved how phenylethanoid glycosides (1–3) reach brain.”  The glycosides were discovered in the rat brains, but how they reach brain is not investigated.

Response 2: In lines 29, it has been revised as “we further proved phenylethanoid glycosides (13) could reach brain.

As the main finding of this article is detecting CTE in rat brain cells, and to make the article more interesting, a more robust discussion is recommended on why the previous studies failed to observe CTE in brain cells

Response 3: (1) The previous studies of CTE considered that the bioavailability of phenylethanoid glycosides is extremely low. Our research indicates that metal chelation must be taken into account to present true bioavailability.  (2) This study provides an advanced and more sensitive UPLC/MS/MS analysis method to analyze the CTE components in tissues.

In line 42: “US dollars 604 billion in 2010” should be changed to 604 billion US dollars.

Response 4: In line 42, it has been revised as “604 billion US dollars

In line 79, provide a reference for sentence: CTE is closely related with anti-Aβ deposition and metal chelation.

Response 5: In lines 78-79, it has been revised and provided references as “CTE is closely related with anti-Aβ aggregation and deposition [24,25].

In line 83, what do you mean by “rare”? The sentence needs to be reformatted.

Response 6: In line 82, it has been revised as “Most studies were designed for quantitative determination of echinacoside, acteoside, and isoacteoside in herbs or foods by HPLC [26–28], but only two studies were for determination in serum or tissues [29,30].”.

In line 88, should be corrected into “underlying mechanism of CTE against Aβ aggregation”. CTE Aβ aggregation implies that Ab aggregation is present even in presence of CTE.

Response 7: In line 88, it has been revised as “against Aβ aggregation and deposition

In line 102, please clarify if all three compounds were mixed with 50% Methanol or each compound was mixed with 50% Methanol?

Response 8: Line 101-102, it has been revised as “Compounds 13 (5.0 mg) were accurately weighed and mixed with 50% methanol (40 mL).”.

In line 103, according to previous comment, if there were three standard solutions, glycoside should not be plural in “Three phenylethanoid glycosides (13) standard solutions …”

Response 9: Line 103, it has been revised as “solution was.

10. In line 105, what does the word “respectively” correspond to?

Response 10: It has been deleted.

11. The verb in the sentence in Line 108 is missing: The intra-day and inter-day precision (% R.S.D. was from 0.1 to 1.8%) and accuracy (95.6–104.2%) by UPLC for 1–3.
In Figure 2, indicate what the unnamed peak stands for. b

Response 11: Line 107-109, it has been added as “The intra-day and inter-day precision (% R.S.D. was from 0.1 to 1.8%) and accuracy (95.6–104.2%) were acceptable by UPLC for 13.”. Figure 2S has been added the unnamed peak (4) as tubuloside A. Origin Figure 2, 3, 4, 5, and 6, respectively, were revised as Figure 2S, 2, 3, 4, and 5, respectively,

12. In Figures 2-5: In all UPLC experiments, the control is missing and must be provided.

Response 12: It has been provided the control data in the Figures 2 (c), 2 (d) and 3 (c).  

13. The quality of the picture in Figure 2 is better to be enhanced. Figure 2, could be moved to supplementary information.

Response 13: Figure 2 has been enhanced the quality and changed as Figure 2S.

14. Line 124: Explanation for Figures. 4A and 4B is missing in the main text. Explain the significance of these chromatograms.

Response 14: Lines 127-128, it has been added as “The phenylethanoid glycosides (13) were observed in hippocampus (Figure 3A) and stratum (Figure 3B).

15. Figure 5, how was obtained? If it is explained in ref 19, then needs to be referenced.

Response 15: Line 142, it has been added the reference [19].

16. Line 143: The sentence following is unclear: The assay was based on that metal chelation changes the retention time of free-form compounds.

Response 16: Lines 145-146, it has been modified as “The assay was based on that metal chelation changes the retention time of free-form compounds in HPLC [34,35].

17. Line 144: The comment “1-3 were prepared 50 μg/ml” should be rephrased.

Response 17: Lines 146-148, it has been modified as “The 50 μg/ml solution of the phenylethanoid glycosides (1-3) were respectively added into a 10 μg/ml solution of copper (Cu), calcium (Ca), magnesium (Mg), zinc (Zn), iron (Fe) or rat serum.”.

18. Line 152: The sentence “The data represented mean ± S.E., n=3. ***P <0.001.” is unclear. What is P?

Response 18: Line 156, “p” is P value (calculated probability). It has been revised as “*** p <0.001 compared with compound alone.”.

19. For Figure 6, explain how the error estimates were acquired. Maybe it is better to insert this answer in the method section.

Response 19: Figure 5 has been revised as “The data represented mean ± S.D., n=3.”.  Lines 323-324, it has been added as “The percentage of metal chelating effect was calculated as below and the data represented mean ± S.D. (n=3)”. in the revised Materials and Methods section.

20. Line 156: The comment: “but failed in large clinical trials targeting Aβ of 2012”, should be rephrased. If 2012, is referred to the year, it should be noted that there have been much more clinical trials since 2012, that they are worth mentioning, although most of them have failed.

Response 20: Lines 169-171,  it has been revised as “Worldwide AD researchers take lots of effort to investigate new anti-AD drugs but failed in large clinical trials targeting Aβ in 2012 such as Pfizer, Lilly, and Johnson & Johnson [9].”.

21. Line 160: “metal chelation hypothesis has become the newest important research direction.” I have to disagree with this comment. The metal chelation in Ab was discovered back more than 2 decades ago. And there have been a lot of studies targeting metal chelation, in the past few years. Many research groups have targeted tau or Ab with antibodies, which are used in the most recent clinical trials.  Hence, the word “newest” should be removed from the sentence.

Response 21: Lines 172-174,  it has been revised as “Therefore, metal ions play an essential role in the pathogenesis of AD, and metal chelation hypothesis has become the important research direction [9, 13–15].”.

22. Line 178: The following remark is not convincing: However, this study shows that 13 would be detected in the hippocampus and striatum of rat brain tissues after CTE oral administration (Figure 5). In Figure 5, the concentration of 2 and 3, especially for 2 are very small. This is in correspondence of the line 177 remark, according to reference 30. The importance of much higher concentration of compound 1, compared to 2 should be emphasized.

Response 22: The content of compound 1 is much higher than 2 and 3. The CTE contents 1 (254 mg/g), 2 (38 mg/g), and 3 (41 mg/g) were quantified by UPLC. (figure 2S). Therefore, it is reasonable to observe that 1 is the highest brain tissue after oral CTE administration.

23. Line 185: Please explain why, or provide a reference for the following comment: The iron metal chelation is the crucial reason leading to misidentification of the very low bioavailability of phenylethanoid glycosides.

Response 23: Lines 200-201, it has been added as “The iron metal chelation may be the crucial reason leading to misidentification of the very low bioavailability of phenylethanoid glycosides.”.

24. Line 187: The sentence: “Figure 6 shows that three phenylethanoid glycosides would have metal chelating activity with iron and serum respectively.” might not be true, as there is possibility that only one or two of the compounds is reacting with iron and serum. Hence, a further study is required in which these compounds are separated and then the experiment is repeated for each compound. The same remark is applicable for the next two sentences (lines 185-190).

Response 24: Figure 5 is performed for each compound. Lines 197-198, it has been revised as “Figure 5 shows that three phenylethanoid glycosides would have metal chelating activity with iron and serum.”, and in lines 321-322, it has been revised as “Each compounds with each metal solution or serum was analyzed by UPLC.”.

25. Line 188: What are you referring by using the word “respectively”?

Response 25:  it has been omitted.

26. Line 250: at 4 (the unit is missing!)

Response 26: Line 262, it has been revised as “4

27. Line 254: explain how you performed “infusing Aβ 1-42 into the cerebral ventricle” How much Aβ was infused?

Response 27: Lines 266-268, it has been revised as “After infusing Aβ 1-42 (300 pmole/day) into the cerebral ventricle for at least two weeks or more, the AD-like rats were given CTE (200 mg/ml) by oral administration on the next day.”.

28. Figure 3 and 4, why the graph for echinacoside is separate from the other two compounds?

Response 28: The total ion chromatograms (TIC) for 13 were carried out using the multiple-reaction monitoring (MRM) mode by UPLC/MS/MS. Compound 1 is presented a single (tR=4.32) at m/z 785.481 [M-H]- and isomer compound 2 and 3 are presented singles (tR=5.28 of 2 and tR=5.85 of 3) at m/z 623.363 [M-H]- in Figure 3. Figure 4 is the same as figure 3.

29. Line 307: rephrase the sentence, I believe you meant the 50 μg/ml solutions of 1-3 were prepared, containing …”

Response 29: Lines 319-321, it has been revised as “UPLC was used to analyze the metal chelating effect using the 50 μg/ml solution of CTE phenylethanoid glycosides 13 respectively containing 10 μg/mL of metal solution of copper, calcium, magnesium, zinc, iron or rat serum.”.

30. Line 308: It is unclear what the word “respectively” stands for.

Response 30: It has been omitted.

Round 2

Reviewer 2 Report

The manuscript has been significantly improved. There are still some minor details that needs to be fixed; such as:

The title: "Mechanism of inhibition of amyloid beta aggregation and deposition of Cistanche tubulosa aqueous extract" No new mechanism of inhibition is provided in this article. The title should be modified.

line 174: "the important research direction" should be changed to "an important research direction". There are other important pathways targeting this issue.

In figure 1, 1-3 should be changed to 1, 2, and 3.

Response 1 is not satisfactory and the sentence is still misleading as no assessment on inhibition is performed in this particular article. I suggest the sentence to be removed. It is possible to be presented in the discussion, and to be linked to the previous published results.

Response 16 is still unclear: {The assay was based on that metal chelation changes the retention time of free-form compounds in HPLC [34,35].}

Response 20 is not satisfactory.Pfizer, Lilly, and Johnson & Johnson” are the names of companies, not the name of the clinical trials.

Response 22 needs modification;

The content of compound 1 {{In what?? Based on what??}} is much higher than 2 and 3.

The second sentence in line 109 could be modified, as the values are from UPLC, it might not necessarily be the true quantity:  The CTE contents (254 mg/g), (38 mg/g), and (41 mg/g) were quantified by UPLC. (figure 2S). could be modified into UPLC quantified the 1, 2, and 3 CTE contents to be 254, 38, and 41 mg/g. 

Also “in” is missing in the last sentence: Therefore, it is reasonable to observe that 1 is the highest in brain tissue after oral CTE administration.

Line 320: “respectively” is not required.

Line 321: s to be removed from: Each compound s

Author Response

Response to the Reviewer Comments

1. The title: "Mechanism of inhibition of amyloid beta aggregation and deposition of Cistanche tubulosa aqueous extract" No new mechanism of inhibition is provided in this article. The title should be modified.

Response: The title has been revised as “Inhibition of amyloid beta aggregation and deposition of Cistanche tubulosa aqueous extract”.

2. Line 174: "the important research direction" should be changed to "an important research direction". There are other important pathways targeting this issue.

Response: Line 169, it has been revised as “an important research direction

3. In figure 1, 1-3 should be changed to 1, 2, and 3.

Response: Line 94, the title of figure 1 has been revised as “Structures of 1, 2, and 3 isolated from C. tubulosa.

4. Response 1 is not satisfactory and the sentence is still misleading as no assessment on inhibition is performed in this particular article. I suggest the sentence to be removed. It is possible to be presented in the discussion, and to be linked to the previous published results.

Response: The sentence has been removed and presented in the discussion as “The in vitro study indicated the active dose of 13 were up to 50 μg/ml [24,25]. But previous studies considered the oral bioavailability of 1 and 2 were very low (0.83% for 1 and 0.12% for 2) in rat [29,30].” in lines 181-183.

5. Response 16 is still unclear: {The assay was based on that metal chelation changes the retention time of free-form compounds in HPLC [34,35].}

Response:  Lines 141-142, it has been revised as In literatures, the determination of the free-from compounds and their metal complex were performed using HPLC [34,35].

6. Response 20 is not satisfactory. “Pfizer, Lilly, and Johnson & Johnson” are the names of companies, not the name of the clinical trials.

Response: Lines 165-166, it had been revised as “Worldwide AD researchers take lots of effort to investigate new anti-AD drugs but failed in several large clinical trials targeting Aβ in 2012 [9].”

7. Response 22 needs modification; The content of compound 1 {{In what?? Based on what??}} is much higher than 2 and 3.

Response: Line 130, it had been revised as “The content of 1 is much higher than 2 and 3 in CTE by UPLC analysis”

8. The second sentence in line 109 could be modified, as the values are from UPLC, it might not necessarily be the true quantity:  The CTE contents (254 mg/g), (38 mg/g), and (41 mg/g) were quantified by UPLC. (figure 2S). could be modified into UPLC quantified the 1, 2, and 3 CTE contents to be 254, 38, and 41 mg/g. 

Response: Line 105-106, it had been modified as “UPLC quantified the 1, 2, and 3 of CTE contents to be 254, 38, and 41 mg/g, respectively (Figure 2S).”.

9. Also “in” is missing in the last sentence: Therefore, it is reasonable to observe that 1 is the highest in brain tissue after oral CTE administration.

Response: Line 131, it had been added as “Therefore, it is reasonable to observe that 1 is the highest in brain tissue after oral CTE administration.

10. Line 320: “respectively” is not required.

Response: It has been omitted.

11. Line 321: s to be removed from: Each compound s

Response: It has been omitted.